# Using Behavioural Skills Training with Healthcare Staff to Promote Greater Independence for People Living with Dementia: A Randomised Single-Case Experimental Design

**DOI:** 10.3390/bs15070870

**Published:** 2025-06-26

**Authors:** Janette Hanniffy, Michelle E. Kelly

**Affiliations:** 1School of Psychology, University of Galway, H91 TK33 Galway, Ireland; hanniffy.janette@gmail.com; 2Department of Psychology, National College of Ireland, D01 K6W2 Dublin, Ireland

**Keywords:** dementia, prompting, single-case experimental design, activities of daily living, independence, behaviour skills training

## Abstract

Approximately 72% of older adults in residential care have dementia and present with different levels of functioning. People living with dementia (PLwD) may not always be facilitated to independently carry out activities of daily living (ADLs) in care, increasing the likelihood of excess disability. This study incorporated Behavioural Skills Training (BST) to train healthcare staff how to increase opportunities for independence for PLwD by using task analyses and least-to-most (L-M) prompting procedures during ADLs. Three healthcare staff, two female and one male (mean age = 42.67, SD = 16.82), participated in the intervention. The What Works Clearinghouse (WWC) Single-Case Design Technical Documentation guided the study’s design. A randomised single-case experimental (N-of-1) design was employed, using a multiple-baseline design (MBD) across participants (*n* = 3) for three separate ADLs. The dependent variable (DV) was the percentage of correct staff responses when implementing the L-M prompting procedure for each step during ADLs. Visual and statistical analyses demonstrated an increase in the correct use of a task analysis and L-M prompting for all three participants during the intervention compared to the baseline: for ADL1 (assistance to stand), effect sizes were d = 5.39, d = 9.38, and d = 6.79 for the three participants, respectively; for ADL2 (assistance with drinking), effect sizes were d = 3.27, d = 8.55, and d = 3.67; and for ADL3 (assistance to brush teeth), effect sizes were d = 5.99, d = 12.93, and d = 9.39. Maintenance data ranged from 70% to 100% correct responses at follow-up (mean = 93.11% SD = 7.85). Participants successfully generalised skills learned to two new ADLs (PLwD eating a meal and putting on a jumper). BST was demonstrated to be an effective training strategy to increase opportunities for independent responding for PLwD in care environments. The contingencies influencing staff behaviour require attention within the healthcare environment.

## 1. Introduction

Dementia is an acquired decline in one or more cognitive domains ([49]) which presents uniquely for each individual but typically includes behavioural changes ([8]). According to the [83] ([83]), over 57 million of the world’s population have a diagnosis of dementia, with 10 million new cases diagnosed per year. Alzheimer’s disease remains the most common cause of dementia ([83]). Dementia is also recognised as a disability by the United Nations Convention on the Rights of People with Disabilities ([3]), meaning that the human rights and fundamental freedoms of people living with dementia (PLwD) are recognised and protected under the convention. This includes people’s right to access the appropriate psychosocial supports that allow them to live independent, engaged, and fulfilled lives for as long as possible.

A decline in ability to independently perform activities of living (ADLs) is common in dementia patients ([10]), and is associated with an increased risk of hospitalisation and admission into long-term care ([5]). Whilst independence and functional ability are often dependent on disease type and stage, functional decline can also be attributable to “excess disability” ([84]). Excess disability arises when disempowerment and restricted perceptions of what PLwD can do reduce their opportunities to function to the best of their ability ([34]). Supporting PLwD to engage in meaningful daily activities (like self-care or household tasks) improves the non-cognitive symptoms of dementia ([50]), sustains a good quality of life ([26]; [78]), and maintains connectedness ([23]) for those living at home or in long-term care. Up to 72% of individuals in residential care have dementia with different levels of functionality ([61]). The literature on activity engagement in residential care suggests that PLwD experience disempowerment ([29]) and are rarely provided with adequate opportunities to engage independently in ADLs ([75]).

Supporting PLwD’s continued engagement in activities can be challenging. The PLwD may experience a decline in motivation to interact with discriminative stimuli (SD) due to a decrease in the reinforcing contingencies experienced ([71]) even though reinforcement is still available ([4]). The ability of SD to evoke behaviour that once commenced or progressed a behaviour chain decreases in effectiveness ([74]; [19]). This can mean that the ability of PLwD to engage in activities decreases over time. This may even occur for behaviours they enjoy and have in their repertoire, and support is required to facilitate continued engagement as the condition progresses ([4]). Non-socially mediated antecedent interventions, such as wayfinding signs to find the bathroom ([55]), and socially mediated antecedent interventions, such as staff prompting PLwD ([57]), may enhance the disparity and salience of stimuli evoking desired behaviour. These interventions should ideally be present in care setting to promote independence in ADLs and reduce excess disability, although research suggests that this is often not the case ([56]).

ADLs are examples of behaviour chains ([53]) and include mobilising, eating, and personal hygiene ([41]). ADLs are activities that PLwD may have in their repertoire, but, due to cognitive changes, they may be unable to commence or continue the chain of behaviour needed to complete the task independently ([53]). The inability to respond independently creates varying levels of dependency, often reducing the individual’s QoL ([47]; [51]). Research shows that PLwD can be supported to engage in ADLs when they are adapted appropriately, which, in return, can increase QoL, reduce caregiver burden, and improve cognitive and functional outcomes ([21]; [24]; [62]). The methods of presenting the steps of ADLs used by the healthcare worker are essential for promoting independence levels ([64]).

Prompting is a method used to increase independent engagement with activities ([16]; [79]; [85]). Prompting can act as an antecedent intervention delivered verbally or visually by another individual or technology. Prompting evokes a response within the target behaviour chain, resulting in the PLwD interacting with the next step in the activity. Most-to-least prompting is effective for individuals when skills are absent from their repertoire, as it allows for fading prompts within a task when stimulus control is transferred so the individual can complete the task independently ([46]). A least-to-most (L-M) approach may be more appropriate for individuals with the skill in their repertoire to increase their engagement with the task ([16]). L-M prompting also creates effortful learning conditions, which support positive learning outcomes for PLwD ([11]). The delivery of instruction should be simple, positive, and direct ([7]), achieved through gestures, pointing, and verbal instruction. Research on prompting has demonstrated increased activity engagement ([4]), independence in ADLs ([16]), and decreasing occurrence of incontinence ([9]; [39]). While prompting has demonstrated some efficacy in increasing independence for PLWD, the research is somewhat limited. Examining the maintenance and generalisation of prompt use is also required, as continued use of L-M prompting for PLwD could sustain engagement in ADLs for longer ([13]; [33]; [79]; [85]).

To promote increased independence in ADLs for PLwD, healthcare staff could be trained in behavioural intervention techniques such as prompting, fading, shaping, and task analysis ([2]; [78]). [6] ([6]) state that the best way to train healthcare staff to promote independence for PLwD has yet to be identified. Behaviour Skills Training (BST) is a well-researched method of training staff to implement behavioural support ([42]) and has been used to improve staff performance in intervention delivery ([58]; [70]). [22] ([22]) reported a significant positive effect of using BST to teach behavioural procedures to front-line staff compared to a control group. Participants gained and generalised skills quickly when instruction, role-play, modelling, and feedback were combined. BST has been shown to be effective for teaching multiple skills to different groups of trainees ([12]; [1]). [60] ([60]) outlined the six steps for BST when applying this model to staff who work in the care environment (see Methods). In their systematic review, [72] ([72]) identified three essential components to quality research in BST: reporting primary outcomes, generalised outcomes, and the maintenance of outcomes. When examining the effects of interventions in settings with vulnerable populations, research design should also be a primary consideration. The Oxford Centre for Evidence-Based Medicine ranks randomised N-of-1 single-case experimental designs (SCEDs) as Level 1 evidence ([28]). Although SCEDs have long been a central feature of behavioural research, SCEDs have only gained interest within the health ([76]) and rehabilitation sectors ([69]) in more recent years. There is now recognition that SCEDs are especially important to optimise personalised, and person-centred, intervention approaches ([15]).

The current study aimed to teach healthcare staff to increase opportunities for independent responding for PLwD by using task analyses and L-M prompting during ADLs (standing, drinking, and brushing teeth). The intervention effects were assessed using a randomised SCED. Generalisation, maintenance, and social validity were also evaluated.

## 2. Materials and Methods

### 2.1. Participants

The participants were three healthcare assistants employed to work with PLwD in a dementia-specific respite centre (mean age of 42.67, SD = 16.82) with an average of 14.4 years’ experience working there. Participants were required to have a minimum of six months’ experience carrying out care duties for at least 12 h a week in the centre. All participants had completed a Further Education and Training Awards Council (FETAC) Level 5 qualification in Care of the Older Person and attended in-house training (for all staff) that included Dementia Awareness, Behaviours that Challenge, and Stress Management. The study information was shared with all staff via information leaflets and word of mouth. Those interested in participating were provided with full study information leaflets. Only staff who subsequently confirmed their interest in participating and provided informed consent (*n* = 3) attended the BST.

### 2.2. Setting

The study took place in a dementia-specific respite centre. PLwD came to the centre from living in the community with their family or caregiver, availed two weeks of respite care, and returned home again. All training sessions occurred in a large dayroom, and staff observations occurred during daily work routines throughout the respite centre.

### 2.3. Experimental Design

The What Works Clearinghouse (WWC) ([36]) Technical Documentation, which outlines design standards for SCDs, was adhered to. The Single-Case Reporting Guideline in Behavioural Interventions (SCRIBE) was referenced for reporting outcomes ([77]). A randomised single-case experimental MBD across participants was used to measure the effects of BST on the participants’ target behaviours. The dependent variable (DV) was the proportion of correct responses when staff were required to use L-M prompting with a task analysis for supporting ADLs (assistance to stand, assistance to drink, brushing teeth). As per the guidelines for conducting randomised SCEDs, participants were randomly assigned to predetermined baseline and intervention lengths for each ADL (Table 1) ([37]; [45]). The BST intervention was implemented in a staggered manner for each participant using within-case (phase) randomisation procedures, as described by [44] ([44]). The ADLs that participants engaged in with the PLwD occurred during the day within their natural work routine. As per the WWC guidelines, an effect replication was assessed across at least three goals per participant, with at least 3–5 data points per phase (baseline and intervention). The MBD (*n* = 3) was replicated across three ADLs: ADL1: assistance to stand, ADL2: assistance with drinking, and ADL3: assistance to brush teeth. The multiple baselines allowed participants to act as a control for themselves and facilitated the measurement of the BST effect across participants for each task ([27]). Staff behaviour was also evaluated to determine whether the skills learned generalised to other ADLs without direct training and feedback. A social validity questionnaire examined staff experiences with BST and whether they would engage in this type of training in the future.

### 2.4. Measurement

The DV was the percentage of correct responses the participant achieved within the task analysis using L-M prompting during each ADL. This percentage was calculated by giving each step a percentage value. Ten steps in the task analysis would mean 10% per step where L-M prompting was correctly used, and six correct responses would correspond to 60% correct overall ([12]; [1]). The L-M prompting sequence used was verbal prompt → verbal and gestural prompt → model prompt → physical prompt ([17]; [52]), with a three-second time delay between prompts ([82]). That is, if the PLwD did not respond within three seconds of the first level prompt (verbal), the next prompt level was offered. Data were recorded on the following: the consistent use of the L-M prompting in each step of the task analysis; the 3 s delay between prompt levels; the use of a total task chaining method so that all steps of the task analysis were completed while allowing the PLwD to chain behaviours they could perform independently, i.e., without any prompting ([52]). Participants were expected to use social reinforcement through appropriate communication and verbal encouragement.

A step on the task analysis was marked as correct if L-M prompting was used at each step with the 3 s time delay. If the resident chained steps of the task together independently without prompting, each step on the task analysis was marked as correct for the participant (i.e., the participant facilitated fully independent responding from the resident). If the resident did not engage in a step and the participant used L-M prompting, this was marked as correct. A step on the task analysis was marked as incorrect if one of the following occurred: the L-M prompting levels were not used, the 3 s time delay was not allowed for, or the PLwD were not allowed to chain steps independently without prompting. The correct answers were added and a percentage used to represent the number of steps in the task analysis that the participant delivered correctly.

### 2.5. Procedure

#### 2.5.1. Baseline

The researcher used a task analysis data collection sheet (available upon request) to gather baseline data on the participant’s use of L-M prompting during each step of the task when assisting a resident to stand, drink, and brush their teeth. Baseline observations occurred in the natural environment and as part of the participant’s daily routine with a person living with dementia. Participants did not receive the task analysis for baseline data observations; they were asked to carry out the ADL as they usually would with the person they supported, and no instruction or feedback was given at this time ([65]; [70]).

#### 2.5.2. Intervention

The BST used the following six steps: 1. describe the target skill, 2. provide a succinct written description of the skill, 3. demonstrate the target skill, 4. require the trainee to practise the target skill, 5. provide feedback during practice, and 6. repeat steps 4 and 5 to mastery ([60]). Steps 1 and 2 were verbal and written training, steps 3 and 4 were rehearsal training, and steps 5 and 6 were performance-based training ([63]). Steps 1–5 were the intervention, while step 6 was where data was collected on participants performance, i.e., they received feedback for improvement following observation.

Step 1. Describe the Target Skill. Participants learned to implement the target behaviour using the L-M prompting hierarchy during each task analysis step. The target behaviours that participants were required to engage in through the Behaviour Skills Training were as follows: (1) Assisting a person living with dementia to stand, which occurred when a resident sitting in a chair stood up under a participant’s prompting (e.g., verbal suggestion) to move from that chair to another location. This did not include the participant placing their hand on a resident to guide or lift them from the chair as a first action. (2) Assisting a person living with dementia to take a drink, which occurred when a resident asked for and was offered a drink and they accepted and took a drink. This did not include the participant lifting the cup to the resident’s mouth as a first action. (3) Assisting a person living with dementia to brush their teeth, which occurred when the resident brushed their teeth when directed to do so. This did not include the participant brushing the resident’s teeth for them.

Step 2. Provide a Written Description of the Skill. Staff received a written description of their expected responses while undergoing direct observations for each target behaviour ([65]). Participants received an information sheet that included their target behaviour, an operational definition of their target behaviour, the task analysis steps, and a description of the least-to-most prompting.

Step 3. Demonstrate the Target Skill. In a role-play scenario, the researcher demonstrated the L-M prompting for each task analysis step with another staff member, and participants were allowed to ask questions ([12]).

Step 4. The Trainee is Required to Practise the Target Skill. Observations of participants began when they performed the target skill in a role-play scenario where they received feedback and had the opportunity to ask questions ([66]). Data collection observations commenced when participants were observed in their interactions with a person living with dementia using L-M prompting during the ADLs.

Step 5. Provide Feedback During Practice. The researcher adopted the use of an evidence-based protocol for feedback: 1. open with a positive statement, 2. reflect on what was performed correctly, 3. state what was performed incorrectly, 4. state how the incorrectly performed steps need to be performed, 5. allow time for questions, and 6. end on a positive statement ([63]).

Step 6. Repeat Steps 4 and 5 to Mastery. During data collection, the researcher provided corrective feedback to staff after each observation. 

#### 2.5.3. Generalisation

Observations of participants assisting a resident to take a piece of food and putting on a jumper were made under the same conditions as the baseline phase after participants completed the baseline and intervention phases of all three ADLs ([58]). These observations allowed the researcher to assess for generalisation of participant learning to break down activities and apply L-M prompting across ADLs that the intervention did not directly target. One of the ADLs to test for generalisation was purposefully similar to assisting with a drink, and the other was not similar, to assess for skills transfer. Participants were not given a written description of the skill.

Assisting a person to take a piece of food during a meal occurred when the PLwD had their meal at the table and used their utensils to take food into their mouths independently. This did not include the participant bringing a utensil to the resident’s mouth as a first action.

Assisting a person to put on a jumper occurred when the resident chose what to wear, held the jumper the right way up, put their arms and head through, and fixed it to sit on their body independently. This did not include the participant placing the jumper on the PLwD as a first step.

#### 2.5.4. Maintenance

Maintenance observations occurred two weeks after each intervention stage was completed for each ADL. Maintenance sessions were carried out under the same conditions as the baseline phase, where participants were instructed to implement each step of the task analysis with the PLwD, and no further instruction or feedback was given ([67]).

#### 2.5.5. Social Validity

A brief ‘social validity’ questionnaire was given to participants to obtain feedback on their experience with BST in the dementia care setting and the impact the intervention had on their overall care skills during the workday. The questionnaire was anonymous and used eight open-ended questions, including, for example, “Is there anything that you would like to change if this was to become part of staff training?”

### 2.6. Analytic Approach

Visual analysis was used to assess the effects of the independent variable (IV) on the DV to determine if a functional relationship existed. Within- and between-condition analyses examined the trend, level, and stability of the data ([40]). A stability envelope was used to assess stability, where the stability criterion was 80% of the data points falling on or within +/−25% of the median value at baseline, and the same envelope was applied for the intervention phase (see [40]). Parallel lines were drawn above and below the median line, and the distance between the two lines demonstrated the amount of variability permitted for the data to be considered stable ([40]). The percentage of non-overlapping data was calculated by taking the number of intervention points that were greater than the highest baseline data point and dividing this by the total number of intervention points and multiplying by 100 ([68]).

[48] ([48]) suggested that quantitative analysis can be helpful in complementing visual analysis. Similarly, [45] ([45]) state that visual analysis should take precedence, with statistical analysis subsequently conducted to demonstrate probabilistically based conclusions and effect size estimates. Statistical analysis was conducted using the Excel Package of Randomization Tests (ExPRT) programme ([20]). A summary across-cases effect size (d) measure was calculated which was the simple average of the individual case ds.

## 3. Results

### 3.1. Visual Analysis

#### 3.1.1. ADL1: Assistance to Stand

Within-Condition Analysis: Evaluation of the relative and absolute level changes within conditions indicated that performance was improving at baseline and during the intervention for P3 (+10 and +10, respectively, at baseline; +17.5 and +35, respectively, during the intervention) and P1 (+5.5 and +11 at baseline; +10 and +25 during the intervention) and deteriorating at baseline and improving during the intervention for P2 (0 and −10 at baseline; +10 and +20 during the intervention). The split-middle method of trend estimation showed an increasing therapeutic trend at baseline and during the intervention for P3 and P1 and a level trend at baseline and an increasing therapeutic trend during the intervention for P2. A stability envelope was applied to trend lines and showed that the data were stable at baseline for P3 but variable during the intervention, while data were stable in both phases for P1 and P2. P3’s mean accuracy scores increased from 51.67% at baseline (range 55–45%) to 83.33% (range 100–65%) during the intervention, and maintenance was 100%. P1’s mean accuracy scores increased from 36.8% (range 45–35%) at baseline to 80% (range 100–65%) during the intervention, and maintenance was 100%. P2’s mean accuracy scores increased from 40.71% (range 45–35%) at baseline to 82% (range 85–65%) during the intervention, and maintenance was 75%.

Between-Condition Analysis: Evaluation of the behaviour change between conditions indicated improvements for the intervention relative to the baseline for all three participants. For P2, a change in performance across conditions meant a change from a level, deteriorating trend at baseline to an accelerating, improving trend during the intervention. For P3 and P1, performance was accelerating and improving across both conditions. All level change measures indicated a positive improving behaviour change across conditions for all participants. The relative level change, absolute level change, mean level change, and median level change were calculated to demonstrate between-condition effects: P3’s scores were +20, +10, +30, and +31.13, respectively; P1’s scores were +34.5, +30, +40, and +43.2, respectively; and P2’s scores were +26, +30, +30, and +36.29, respectively. Figure 1, Figure 2 and Figure 3 provide a visual display of the data. No overlapping data (Table 2) was present for any participants for ADL1 (PND = 100%), indicating that the intervention was very effective ([68]). Tables displaying the full set of visual analysis data can be accessed via the Appendix A.

#### 3.1.2. ADL2: Assistance with Drinking

Within-Condition Analysis: Evaluation of the relative and absolute level changes within conditions indicated that performance was level at baseline and improving during the intervention for P1 (0 and 0, respectively, at baseline; +10 and +30, respectively, during the intervention) and P3 (0 and 0 at baseline; +20 and +20 during the intervention), and performance was deteriorating at baseline and improving during the intervention for P2 (−10 and −10 at baseline; +25 and +30 during the intervention). The split-middle method of trend estimation indicated that there was a level, stable trend at baseline and an accelerating therapeutic trend during the intervention for P1 and P3 and a decelerating contra-therapeutic trend at baseline and increasing therapeutic trend during the intervention for P2. A stability envelope was applied to trend lines and showed that the data were stable at baseline and during the intervention for all three participants. P1’s mean accuracy scores increased from 73.33% at baseline (range 70−80%) to 92.22% (range 90–100%) during the intervention, and maintenance was 90%. P3’s mean accuracy scores increased from 36% (range 30–40%) at baseline to 82.86% (range 70–100%) during the intervention, and maintenance was 100%. P2’s mean accuracy scores increased from 60% (range 50–70%) at baseline to 90% (range 70–100%) during the intervention, and maintenance was 90%.

Between-Condition Analysis: Evaluation of the behaviour change between conditions indicated improvements for the intervention relative to the baseline for all three participants. For P1 and P3, a change in performance across conditions meant a change from a stable, level trend at baseline to an accelerating, improving trend during the intervention. For P2, performance went from a decelerating, deteriorating trend at baseline to an accelerating, improving trend during the intervention. The relative level change, absolute level change, mean level change, and median level change were calculated to demonstrate between-condition effects: P1’s scores were +20, 0, +20, and +18.89, respectively; P3’s scores were +35, +40, +40, and +46.86, respectively; and P2’s scores were +25, +20, +40, and +30, respectively (see Figure 2). P1 had one overlapping data point (PND = 88.8%), P3 had no overlapping data points (PND = 100%), and P2 had one overlapping data point (PND = 80%). The PND scores indicate that the intervention was very effective (Table 2).

#### 3.1.3. ADL3: Assistance to Brush Teeth

Within-Condition Analysis: Evaluation of the relative and absolute level changes within conditions indicated that performance was accelerating and improving at baseline and during the intervention for P3 (+12 and +12, respectively, at baseline; +17 and +22, respectively, during the intervention); level and improving at baseline and accelerating and improving during the intervention for P2 (0 and +6 at baseline; +5.5 and +11 during the intervention); and level and stable at baseline and during the intervention for P3 (relative and absolute level changes were zero within each condition). The split-middle method of trend estimation indicated that there was an accelerating therapeutic trend at baseline and during the intervention for P3; a level therapeutic trend at baseline and an accelerating therapeutic trend during the intervention for P2; and a level trend at baseline and during the intervention for P1. A stability envelope was applied to trend lines and showed that the data were stable at baseline and during the intervention for all three participants. P3’s mean accuracy scores increased from 52% (range 44–56%) at baseline to 94% (range 78–100%) during the intervention, and maintenance was 94%. P2’s mean accuracy scores increased from 53% (range 50–56%) at baseline to 98% (range 89–100%) during the intervention, and maintenance was 100%. P1’s mean accuracy scores increased from 77% (range 72–78%) at baseline to 100% (range 100–100%) during the intervention, and maintenance was 89%.

Between-Condition Analysis: Evaluation of the behaviour change between conditions indicated better performance during the intervention phase relative to the baseline for all three participants. For P3, data were accelerating and improving in the baseline and intervention phases. For P2, a change in performance across conditions meant a change from a level, improving trend at baseline to an accelerating, improving trend during the intervention. For P1, data were level with a stable trend both at baseline and during the intervention. The relative level change, absolute level change, mean level change, and median level change were calculated to demonstrate between-condition effects: P3’s scores were +27, +22, +44, and +41.5, respectively; P2’s scores were +41.5, +33, +47, and +44.8, respectively; and P1’s scores were +22, +22, +22, and +23, respectively (see Figure 3). All participants had no overlapping data for ADL3 (PND = 100%), suggesting that the intervention was very effective.

### 3.2. Statistical Analysis

Within the ExPRT package, the Wampold–Worsham model (within-case comparison) procedure was selected. Each case was assigned to a single predetermined intervention start point, and the cases were randomly assigned to stagger positions within the multiple-baseline design (see [20]). Outputs from ExPRT included summary data, including plotted data, individual case means and standard deviations, and effect sizes. We intended to include *p*-values for A-to-B-phase mean differences, but the study was statistically underpowered and unable to produce conventional *p*-values. Instead, we report the across-cases mean B-A difference (raw score) and a summary across-cases d measure, which is the simple average of the individual case ds (i.e., average standardised effect size). [59]’s ([59]) NAP index, is also reported for each case. NAP indicates the proportion of A- and B-phase observation outcomes that are non-overlapping. ExPRT’s individual-case NAP indices are rescaled from Parker and Vannest’s original NAP measure (NAPPV), which ranges from 0.50 to 1.00, so that the rescaled NAP measure (NAPR) ranges from 0.00 to 1.00.

For behaviour 1 (assistance to stand), the across-cases mean B-A difference was 36.86 raw-score units. The average effect size was d = 5.85 (i.e., the average mean increase between Phase A and Phase B amounted to almost six A-phase standard deviations). The average effect size of NAP was 1.00. For behaviour 2 (assistance with drinking), the across-cases mean B-A difference was 31.92 raw-score units, the average effect size was d = 5.17, and the average effect size of NAP was 0.932. For behaviour 3 (assistance to brush teeth), the across-cases mean B-A difference was 36.43 raw-score units, the average effect size was d = 9.44, and the average effect size of NAP was 1.00.

### 3.3. Interobserver Agreement

Two healthcare staff members (not otherwise involved with the research) were trained to collect interobserver agreement (IOA) data. Data were collected simultaneously in different locations in the same room. IOA data was collected once per condition (baseline and intervention) for each participant during the three ADLs. IOA was calculated using the exact agreement IOA: the number of exact agreement trials/the total number of trials × 100. The number of ‘trials’ per ADL depended on the number of steps in the task analysis. Staff used the task analysis data collection sheet to gather this data. The overall average IOA was 94%, with scores ranging from 77.8% to 100%.

### 3.4. Generalisation

Generalisation probes for assisting an individual with eating were 90% for participant 1, 100% for participant 2, and 90% for participant 3. Helping a person living with dementia put on a jumper was 80% for P1, 90% for P2, and 70% for P3.

### 3.5. Social Validity

The responses to the social validity questionnaire are displayed in Table 3. Overall, participants identified an increased awareness of how to promote higher levels of independence and a willingness to use this in activities outside of what was directly taught but also recognised that time was an issue. Participants stated that they would recommend the training to others but exhibited mixed results for wanting to engage in the same type of training in the future.

## 4. Discussion

This study incorporated BST as an intervention to train healthcare staff in a dementia care facility to increase opportunities for independent responding for PLwD during ADLs. Specifically, staff were trained on how to use a task analysis to identify the steps in a chain of behaviours required to complete ADLs, and to use L-M prompting procedures to ensure that PLwD were given the opportunity to complete ADLs as independently as possible. The environment was still supportive, in that staff were available to guide PLwD as needed, but, instead of staff defaulting to completing ADLs for PLwD (e.g., full physical prompting), the staff were encouraged to assume that residents could engage in their ADLs independently and then only offered graded prompts as needed. The visual within- and between-condition analysis demonstrated improvements in the intervention phase relative to the baseline phase for the three participants and for each ADL, with level change measures demonstrating relative, absolute, mean, and median improvements between phases. For most participants, there was an abrupt and immediate intervention effect observed with no overlapping data points; however, for ADL2, there was overlapping data for P1 and for P2. Overall, visual analysis of the data suggested a positive behaviour change in the expected direction. The visual analysis was supported by statistical data from the ExPRT package of randomisation tests ([45]) which showed positive pre–post-intervention changes for each participant across the three ADLs, with effect sizes ranging from 3.97 to 12.93 (average effect size of 7.10).

Maintenance data suggested that the intervention effects were maintained at a 2-week follow-up, with scores ranging from 75% to 100%. The percentage of correct responses in the generalisation probes ranged from 80% to 100%. The generalisation of the skills learned during training courses is an important consideration for those who provide training to healthcare staff to promote independence for those in their care. In our study, we incorporated the intervention using an MBD across participants, and we repeated this training and testing across three separate ADLs. The advantage of this was to ensure that we programmed for generalisation, by promoting skill development across multiple exemplars and contexts ([18]; [25]). An examination of the data suggests that training on two to three separate ADLs may be required for staff to be able to generalise this skill to other ADLs. The data varied somewhat across participants, but we can see from P1’s data that, in ADL2 and ADL3, there may have been learning carryover effects from ADL1 (either that or the participant was already facilitating independent responding for these ADLs). However, for P2 and P3, repeated exposure to the BST across three ADLs seemed to be necessary/sufficient in order to promote generalisation to other ADLs. Our findings have implications for those planning staff training, specifically, that training in varied contexts or using different exemplars may be optimal for promoting maintenance and generalisation of the learned skill. This is in line with previous research ([18]; [25]). Future studies should examine specific strategies to ensure maintenance and generalisation of trained skills in healthcare settings.

The strength of a multiple-baseline design is that threats to internal validity can be controlled for, and the MBD can determine if a functional (casual) relationship exists between the intervention and outcome ([73]). A functional relationship is typically inferred if the data (visually) demonstrate experimental control through prediction, replication, and verification; specifically, the intervention produces a change in the outcome variable in a precise and reliable fashion. [73] ([73]) explain that, in (traditional non-randomised) MBD research, baseline stability is important for prediction; a stable baseline predicts that the data path will continue as observed without the intervention. Verification happens when there is no change in the data trends in other staggered baseline tiers also not subjected to the intervention, and replication is observed when an intervention effect occurs across multiple tiers ([73]). However, to achieve case randomisation for a randomised MBD, cases are randomly assigned to *predetermined* intervention start points/baseline lengths ([45]). In practical terms, this meant that baseline stability did not determine when our intervention commenced; intervention commencement was predetermined. The impact of the predetermined baseline lengths can be seen in our data, where there are ascending trends evident during the baseline phases for ADLs 1 and 3. In a traditional non-randomised design, researchers continue to gather baseline data until a stable pattern emerges, but this response-guided approach does not occur with case randomisation and start point randomisation. Some may consider this a limitation to the current study (e.g., see [32]); however, the more recent literature provides important insights into the importance of randomisation for reducing/controlling for threats to internal validity ([38]; [76]), removing researcher bias ([45]), and facilitating the use of randomisation statistical tests ([14]). As [30] ([30]) aptly stated, “randomization is the hallmark of scientifically credible intervention research” (p. 11). Overall, the benefits of randomisation in SCED appear to outweigh the caveats, and SCED researchers should continue to purposefully explore the utility of randomised designs in their work.

Visual analysis of SCED data has been commonplace for over 60 years ([38]). More recently, researchers have been advocating for the use of statistical analysis to supplement visual outcome data ([30]; [45]). Another advantage of using randomised designs in single-case research is to facilitate the use of appropriate randomisation tests, like those described by [20] ([20]). While a strength of our study was the incorporation of randomisation and statistical data, a limitation of our study was the fact that, with only three participants and case randomisation, the study was underpowered and ExPRT was unable to return meaningful *p*-values for the within-case comparisons. The [80] ([80]) model was used in ExPRT as this was deemed appropriate for single fixed intervention start points in a MBD ([20]; [45]). If, however, we had incorporated start point randomisation between two (or more) possible intervention start points, the statistical power of the design would have increased. With start point randomisation, a statistical model like the [35] ([35]) model can be adopted ([20]), which is more powerful with respect to detecting immediate abrupt intervention effects ([43]), as seen with our data. Future studies should consider the use of multiple randomisation procedures (see [30]) to improve credibility and statistical power. Using both case and start point randomisation is a good starting point at least.

With a view to further improving the quality of this research, we designed the study in accordance with the recommendations of the WWC guidelines ([36], [37]). There are four criteria used to determine whether a study design meets design standards, either with or without reservations. In our case, first, we systematically manipulated an IV (BST) which met design standards. Second, we measured the outcome variable in a systematic manner and calculated IOA data on each case for each outcome variable. IOA data should be calculated for more than 20% of the data points within each condition to fully meet the design standards. For our study, IOA was gathered once per condition, meaning that conditions with up to five data points had sufficient IOA data gathered to meet the design standards. However, the IOA gathered for conditions with 6+ data points did not meet the design standards. Third, our study included at least three attempts to demonstrate an intervention effect at different time points. MBDs like ours with at least three baseline conditions meet the design standards. A further strength of our study was that we also replicated the MBD across participants for three separate ADLs. Finally, our MBD included a minimum of six phases (at least three A and three B phases) with at least three data points per phase. This aspect of the design met the standards with reservations (to fully meet the design standards, a minimum of five data points per phase is required). While we found the design standards generally achievable, there were some challenges, particularly in taking a minimum of five data points in the first baseline phase (which would have meant quite an extended baseline, e.g., for the third baseline phase) and in the availability of staff for gathering IOA data. To gather IOA data for this study, the researcher needed to train other staff members who were not familiar with task analysis and L-M prompting to be able to identify ‘correct/incorrect’ responding. The staff did not wish to participate in the research but did attend training to support the researcher with gathering IOA data. While not impossible, this can create tensions and barriers within already busy work environments. More detailed concerns/criticisms of the WWC guidelines have been noted elsewhere ([81]). Despite this, the guidelines offer clear and useful information to support researchers to implement standardised approaches and improve overall design quality.

While our research demonstrates that training can transform staff behaviour to support PLwD to reach their potential functioning, the contingencies that maintain staff behaviour of disempowerment require consideration. The literature is clear that creating opportunities for independent engagement is crucial for PLwD ([84]), yet care settings often do not provide those opportunities ([75]). The creation of dependency for PLwD can directly result from a cycle of positive reinforcement on the healthcare worker created by the workplace environment. Healthcare staff who complete high volumes of tasks in short time frames can receive praise from management and be perceived as hard-working. Those who take longer may receive positive punishment, decreasing their actions to promote independence as they require more time with the PLwD ([6]). If independence levels of PLwD are contingent on staff delivery of the activity, and staff delivery of the activity is contingent on the reinforcing and punishing agents within the work environment, it is essential to examine where the responsibility of promoting quality care lies ([31]). A strong link exists between the maintained outcomes in behavioural gerontology and organisational behavioural management ([31]).

Due to the nature of the respite centre and the high number of admission/discharges of residents, participants implemented the intervention with different PLwD. No data was recorded on the impact of the prompting procedure on the PLwD and whether their independence increased over time. Due to the project’s duration, time restrictions were a limitation. Longer maintenance phases would give better data on skill maintenance. The possibility of observer reactivity of care staff proved to be a limitation which may be addressed where onsite cameras are in place or with video observations. A strength of the current study is the randomised design. Although some disagree about the necessity for randomisation within SCDs ([45]), particularly where more traditional behavioural studies may only begin an intervention on a stable baseline (as discussed above), others suggest that, if SCDs are to be comparable to RCTs as Level 1 evidence, randomisation is required to reduce threats to internal validity and improve causal inference ([37]; [38]; [45]).

## 5. Conclusions

Creating environments that support activity engagement and independence in ADLs has important implications for reducing excess disability for PLwD and encouraging a rights-based approach in line with the UNCRPD. This research adds important insights into how staff can create more independent environments for those PLwD in their care. We suggest that training on task breakdown (task analysis), task presentation, and L-M prompting should be part of staff continuous education, and this should ideally be followed with supervision and positive feedback to increase staff implementation of supportive practices in their work. BST may be more socially valid for staff if it is part of their training and work routine, and if it is supplemented with supportive feedback. BST could also be used to teach other important skills relevant for positive dementia care environments, such as training staff to conduct functional assessments of the behavioural symptoms of dementia ([54]). Future research should gather data on the responses and feedback of PLwD where BST is used to train healthcare staff to support increased opportunities for independence. It would also be interesting to examine the effects of increased independence on behaviours such as refusal during personal care. Time and workload requirements for care staff may pose barriers to supportive intervention implementation ([2]); ensuring staff are afforded the time and space to support PLwD to be independent is crucial. In future research, social validity assessment should focus on feedback from PLwD on the impact that staff training has on their lived experience. Social validity assessment could also focus on staff experience of BST and compare BST to other traditional training approaches. Openness to learning ‘new’ ways of doing things may require a review of values and valued actions within the healthcare environment.

## Figures and Tables

**Figure 1 behavsci-15-00870-f001:**
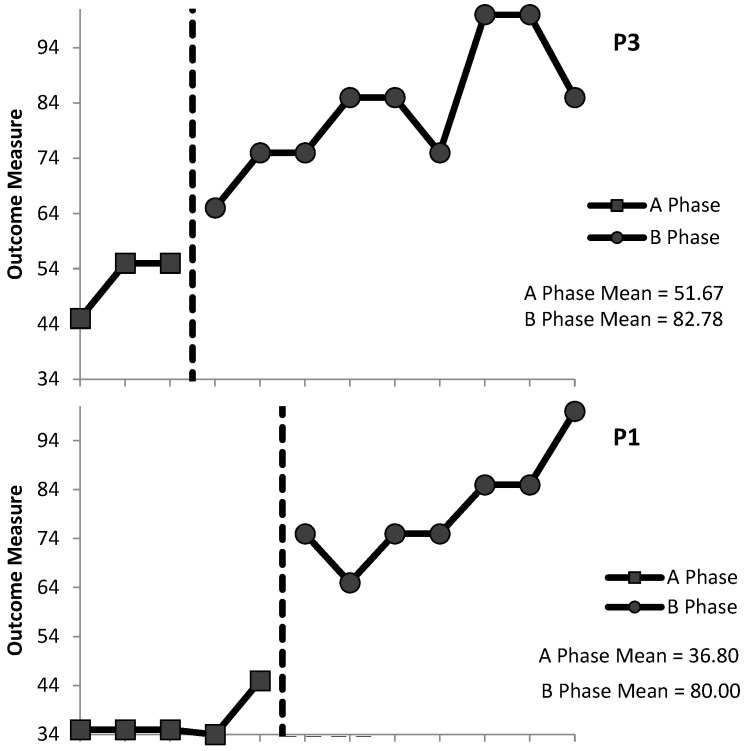
Percentage of correct staff responses (P3, P1, and P2) during Behaviour Skills Training (BST) for activity of daily living 1 (ADL1) (assistance to stand) for the baseline (A) phase and the intervention (B) phases. Mean scores are included under the figure legend. Graphs were created by the ExPRT programme.

**Figure 2 behavsci-15-00870-f002:**
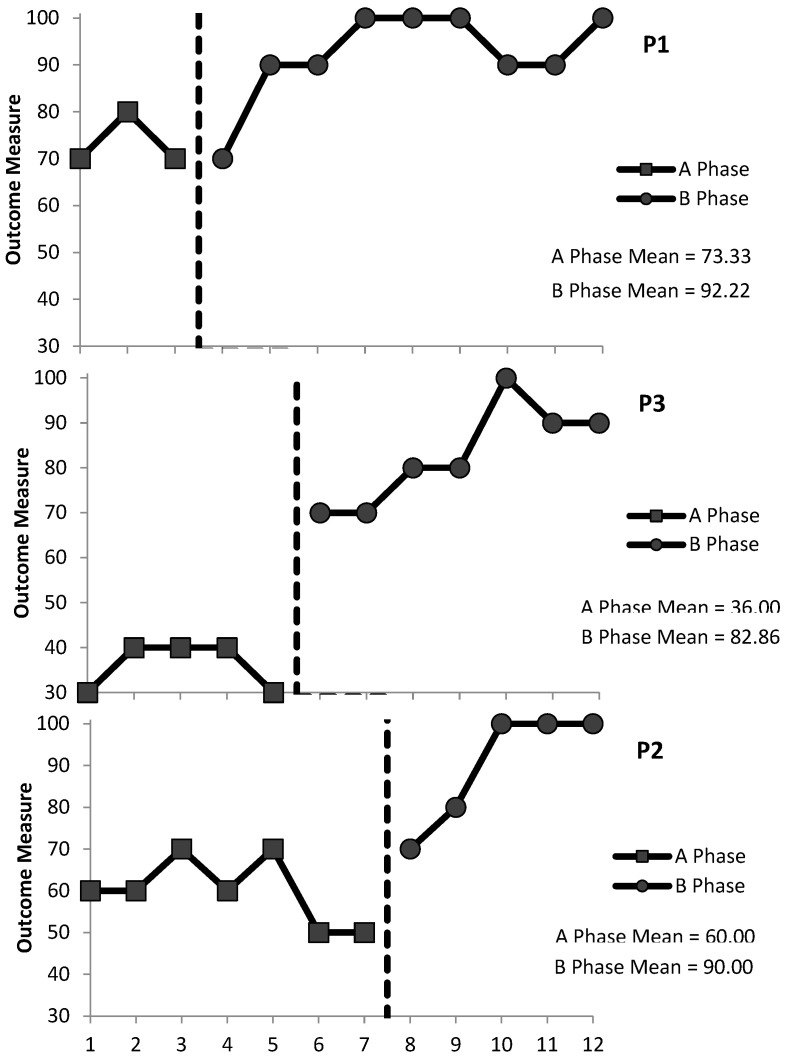
Percentage of correct staff responses (P1, P3, and P2) during BST for ADL2 (assistance with drinking) for the baseline (A) phase and the intervention (B) phases. Mean scores are included under the figure legend.

**Figure 3 behavsci-15-00870-f003:**
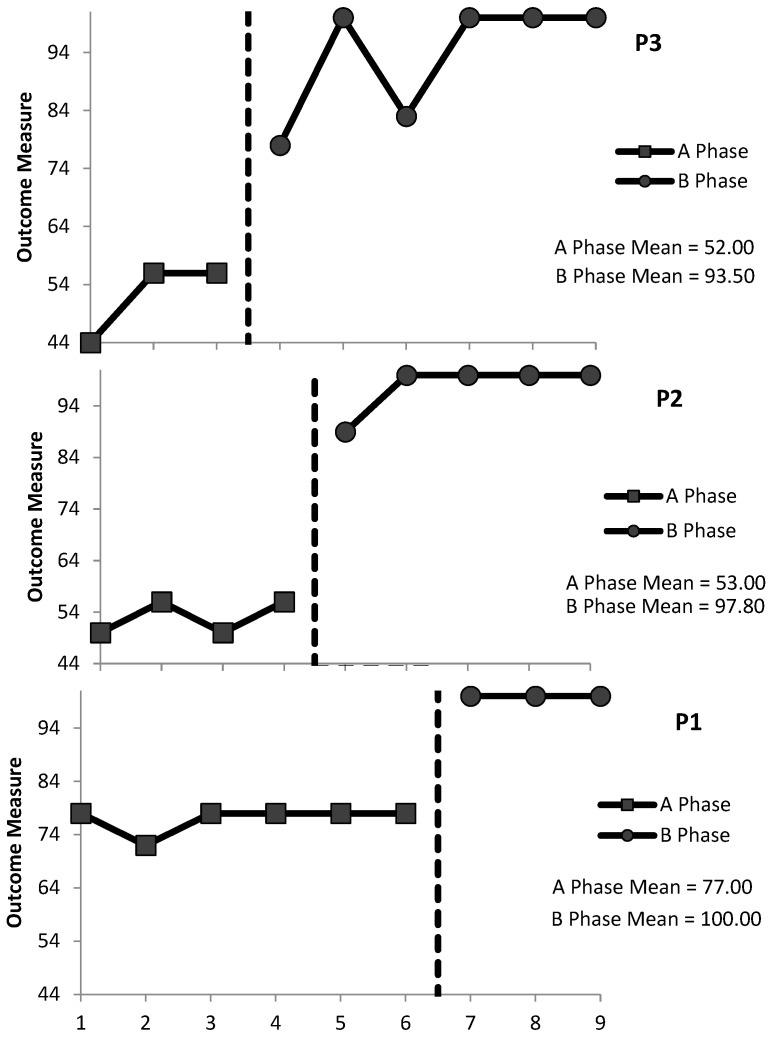
Percentage of correct staff responses (P3, P2, and P1) during BST for ADL3 (assistance to brush teeth) for the baseline (A) phase and the intervention (B) phases. Mean scores are included under the figure legend.

**Table 1 behavsci-15-00870-t001:** Experimental design for each activity of daily living (ADL) displaying the number of baseline and intervention data points per phase.

ADL1: Assistance to Stand			
Participant	Baseline	Intervention	Design
3	3	9	AAABBBBBBBBB
1	5	7	AAAAABBBBBBB
2	7	5	AAAAAAABBBBB
ADL2: Assistance to Drink			
Participant	Baseline	Intervention	Design
1	3	9	AAABBBBBBBBB
3	5	7	AAAAABBBBBBB
2	7	5	AAAAAAABBBBB
ADL3: Assistance to Brush Teeth			
Participant	Baseline	Intervention	Design
3	3	6	AAABBBBBB
2	4	5	AAAABBBBB
1	6	3	AAAAAABBB

**Table 2 behavsci-15-00870-t002:** The percentage of non-overlapping data (PND) observed between baseline and intervention conditions for *n* = 3 participants across three ADLs.

Percentage of Non-Overlapping Data			
	ADL1	ADL2	ADL3
Participant 1	100	88.8	100
Participant 2	100	80	100
Participant 3	100	100	100
Overall PND	100	89.6	100
Effect Level ^1^	Very effective	Very effective	Very effective

^1^ Based on [68]’s ([68]) quality judgement scores.

**Table 3 behavsci-15-00870-t003:** Responses to the social validity questionnaire.

	Social Validity Question	Staff Responses
1.	How do you feel participating in the behaviour skills training has impacted your interactions with individuals with dementia?	I can promote more independence when helping It was fine I feel like I can break small things down; I didn’t think about this before
2.	Do you feel the skills you have learnt benefit individuals with dementia? Why?	Yes, they can do more than I thought Yes, made me think about my actions Yes, but it will take longer
3.	Do you feel you can apply the skills you gained during the study to your work outside the activities of living that were directly taught? Where do you think this applies?	I will try Yes Yes, but it might take practice
4.	Do you feel that this exercise increased your job satisfaction when working directly with individuals with dementia?	Yes Not really No
5.	Would you participate in a similar type of training in the future?	Yes Probably not Not sure
6.	Would you recommend this training to other healthcare staff members? Why?	Yes Yes, some staff need to slow down and give patients more time Yes, but being watched was strange
7.	Do you feel more likely to promote independence within your work practice? Why?	Yes, more aware If I have time I will try
8.	Is there anything that you would like to change if this was to become part of staff training	No No No

## Data Availability

The original data presented in the study are openly available on ResearchGate at DOI: 10.13140/RG.2.2.29452.42882.

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
