# Peer review of "Using Behavioural Skills Training with Healthcare Staff to Promote Greater Independence for People Living with Dementia: A Randomised Single-Case Experimental Design"

_behavsci, 2025, doi:10.3390/bs15070870_

Round 1

Reviewer 1 Report

Comments and Suggestions for Authors

Thank you for the request to review this manuscript. The authors conducted a a single-case experimental interventional study that evaluated behavioral skills training for healthcare staff at a dementia facility as an intervention for promoting greater opportunities for independence for People living with dementia. 

The manuscript is very well-written and organized. I had only minor changes recommended. 

1) The introduction is too long to facilitate readability. I would suggest to shorten the introduction as the authors see fit. Focus on introducing the main topic of the study. 

2) Lines 164-176 might be better suited for the methods section rather than introduction. 

3) Line 36: revise definition of dementia. Dementia is a syndrome rather than a disorder in itself. I would define it as “acquired decline in one or more cognitive domains” as the authors described also. However, this is just a suggestion and not a must. 

Author Response

R1:

  • The introduction is too long to facilitate readability. I would suggest to shorten the introduction as the authors see fit. Focus on introducing the main topic of the study. 

Thank you for your suggestion. The introduction was revised/shortened, while still attempting to maintain the positive qualities highlighted by R2. 1803 words revised down to 1287.

  • Lines 164-176 might be better suited for the methods section rather than introduction. 

Yes, we agree. Moved this to add to the design section (deleted some repetitive content). Highlighted in yellow.

  • Line 36: revise definition of dementia. Dementia is a syndrome rather than a disorder in itself. I would define it as “acquired decline in one or more cognitive domains” as the authors described also. However, this is just a suggestion and not a must. 

Yes, agreed! We revised the definition as suggested and added updated information from the WHO 2025. Highlighted in yellow.

Reviewer 2 Report

Comments and Suggestions for Authors

It is a very well written work in which the ideas flow coherently throughout the manuscript and which complies with all the sections required for its completion.

In the Introduction section, the state of the art is masterfully presented with numerous bibliographical references about the limitations that PLwD may develop in ADLs due to the progress of their cognitive impairment and the benefits of increasing their functional abilities and creating positive repercussions on both their cognitive performance and their quality of life in general.

Material and methods: Within this section, it is suggested to the authors that, in the subsection ‘participants’, it could be further clarified how the three caregivers were selected for the BST. In particular, whether this training was offered to the entire staff, etc.

I think that the sections on Experimental design, Measurement and Procedure are very correct and well detailed. The steps of the intervention are very well specified, which is very valuable information for anyone who wants to carry out an intervention of similar characteristics.

The Results section is very well presented and very complete as they are offered in different modalities such as Visual Analysis, Statistical analysis, Interobserver agreement, Generalization and Social Validity. Likewise, the discussion section is also exhaustive, interpreting the results of the study in the light of previous research.

The conclusions presented are supported by the results and thanks to them both the generalization and maintenance of the intervention confirm the interest of applying this type of training to the staff to learn how to use a task analyzing to identify the steps in a chain of behaviors required to complete ADLs and to use L-M prompting procedures to ensures that PLwD were given the opportunity to complete ADLs independently as possible. 

Author Response

Thank you so much for the positive and encouraging comments.

Suggested revision:

Material and methods: Within this section, it is suggested to the authors that, in the subsection ‘participants’, it could be further clarified how the three caregivers were selected for the BST. In particular, whether this training was offered to the entire staff, etc.

Response: The following information was added to the participants section: The study information was shared with all staff via information leaflets and word-of-mouth. Those interested in participating were provided with full study information leaflets. Only staff who subsequently confirmed their interest in participating and provided informed consent (n=3) attended the BST training.

Notably, the care centre was small, and staff time was very limited. Only those few who were interested in the study had time allocated to attend the additional training.